

# New insights into the mechanism of Schiff base synthesis from aromatic amines in the absence of acid catalyst or polar solvents

Pedro J. Silva[1,2]

[1] FP-ENAS/Fac. de Ciências da Saúde, Universidade Fernando Pessoa, Porto, Portugal
[2] UCIBIO@REQUIMTE, BioSIM, Departamento de Biomedicina, Faculdade de Medicina, Universidade do Porto, Porto, Portugal

## ABSTRACT

Extensive computational studies of the imine synthesis from amines and aldehydes in water have shown that the large-scale structure of water is needed to afford appropriate charge delocalization and enable sufficient transition state stabilization. These insights cannot, however, be applied to the understanding of the reaction pathway in apolar solvents due their inability to form extensive hydrogen-bonding networks. In this work, we perform the first computational studies of this reaction in nonpolar conditions. This density-functional study of the reaction of benzaldehyde with four closely related aromatic amines (aniline, *o*-toluidine, *m*-toluidine and *p*-toluidine) shows that, although an additional molecule of amine may provide some stabilization of the first transition state even in the absence of a hydrogen bonding network, this is insufficient to achieve high reaction rates. Our computations also show that when an extra proton is added to the spectator amine, the activation energies become so low that even picomolar amounts of protonated base are enough to achieve realistic rates. Additional computations show that those minute amounts of protonated base may be obtained under reaction conditions without the addition of extraneous acid through the auto-protolysis of the amines themselves. To our knowledge, this is the first report of a role for the auto-protolysis of anilines in their extensive reactional repertoire.

Corresponding author
Pedro J. Silva, pedros@ufp.edu.pt

## INTRODUCTION

Imines can be readily synthesized through the reversible reaction of amines with aldehydes. This reaction proceeds through an addition step which forms a carbinolamine intermediate, which is then dehydrated to the imine in the rate-determining step. The released water is usually removed from the system to shift the equilibrium towards the products. The reaction rate is quite sensitive to pH: moderate amounts of acid greatly accelerate it (*Santerre, Hansrote & Crowell, 1958*), but excess acid prevents it (*Jencks, 1959* and references therein). The decrease in reaction rate at very low pH is due to the protonation of the amine, which renders it unable to directly attack the carbonyl, whereas protonation of the carbinolamine is required to achieve high rates of dehydration (*Jencks, 1964*) (Fig. 1). Acid catalysis has also been postulated to proceed through

**Figure 1 Possible pathways for the reaction of aldehydes with amines.**

protonation of the carbonyl group, rendering it more susceptible to nucleophilic attack by the amine (*Hammett, 1940*). The reaction may also take place in the absence of acid catalysis (*Law, 1912*; *Campbell et al., 1948*; *Crowell & Peck, 1953*), especially with primary amines.

Computational studies of this reaction have shown that in the absence of charge stabilization by solvent the activation energies of the formation of the carbinolamine (*Hall & Smith, 1998*; *Ding, Cui & Li, 2015*; *Ćmikiewicz, Gordon & Berski, 2018*) are prohibitively high (above 25 kcal·mol$^{-1}$) and the activation energies of its dehydration to imine (*Hall & Smith, 1998*; *Ćmikiewicz, Gordon & Berski, 2018*) are even higher (between 45 and 55 kcal·mol$^{-1}$). Incorporation of one (*Hall & Smith, 1998*; *Ding, Cui & Li, 2015*) or two (*Hall & Smith, 1998*) water molecules as proton transfer assistants greatly facilitates the formation of the carbinolamine by decreasing the activation energy to 8–16 kcal·mol$^{-1}$ but still affords large barriers incompatible with room-temperature reaction (26.7 kcal·mol$^{-1}$) for the dehydration step (*Hall & Smith, 1998*). Realistic barriers are, however, obtained when a large number of explicit water molecules (from 9 to 29) are included in the model (*Solís-Calero et al., 2012*), enabling extensive stabilization of the nascent charges present in the transition state of the dehydration step. Since so far all the computational work on this reaction has been performed on systems including only water as solvent, the aforementioned insights cannot be directly applied to reactions in nonpolar or aprotic solvents, such as the condensation of benzaldehyde with aniline (or toluidines), which is experimentally observed to proceed readily and exothermically in the absence of an acid catalyst (*Law, 1912*; *Campbell et al., 1948*; *Crowell & Peck, 1953*) or protic solvents. In the computational study described in the present manuscript we found, for the first time, a reaction pathway that affords realistic reaction barrier in the

absence of hydrogen-bonding stabilization by protic solvent molecules, and consequently an explanation of how this classic reaction can proceed in nonpolar solvents.

## Computational methods

The geometries of putative intermediates and transition states in the reaction mechanism were optimized using the widely used PBE0 functional (*Ernzerhof & Scuseria, 1999*; *Adamo & Barone, 1999*), which we have earlier shown to be a very good choice for the description of mechanisms involving the protonation or deprotonation of ketones and amines (*Silva & Ramos, 2011*). All geometry optimizations were performed with the Firefly (*Granovsky, 2013*) quantum chemistry package, which is partially based on the GAMESS (US) (*Schmidt et al., 1993*) code, using autogenerated delocalized coordinates (*Baker, Kessi & Delley, 1996*). In geometry optimizations, the aug-pcseg-1 basis set (*Jensen, 2014*) was used for heavy atoms and the pcseg-1 basis set was used for hydrogen. Zero-point and thermal effects on the free energies at 298.15 K were computed at the optimized geometries. DFT energies of the optimized geometries were then computed using the pcseg-2 basis set (*Jensen, 2014*), which is expected to be close to the complete basis set limit for DFT. The double-hybrid functional DSD-BLYP (*Kozuch, Gruzman & Martin, 2010*) supplemented with DFT-D3-BJ corrections (*Grimme, Ehrlich & Goerigk, 2011*) was chosen for these single-point energies due to its superlative performance in the computation of total energies vs. the highest quality benchmarks available (*Goerigk & Grimme, 2011*; *Goerigk et al., 2017*). Auto-protolysis constants (pKs) of various amines, ethylene carbonate, acetonitrile, and nitromethane were computed by comparing the energies of separately optimized neutral clusters of each molecule to clusters of the same size which included one single instance of protonated (or deprotonated) molecule. In all cases, intra- and inter-molecular dispersion effects were included in the geometry optimization, frequency calculation, and high-level single point steps using the DFT-D3 formalism with Becke-Johnson damping developed by *Grimme et al. (2010)* and *Grimme, Ehrlich & Goerigk (2011)*. Solvation effects in aniline were computed using the Polarizable Continuum Model (*Tomasi & Persico, 1994*; *Mennucci & Tomasi, 1997*; *Cossi et al., 1998*) implemented in Firefly. Dispersion and repulsion interactions with the continuum solvent were computed using the method developed by *Amovilli & Mennucci (1997)*. To obtain estimates of the stability of the postulated bimolecular complexes, the potential of mean force was obtained through umbrella sampling simulations performed by constraining the distance between the C = O in benzaldehyde (or carbinolamine) and the $NH_3^+$ of anilinium (or N–H in a second carbinolamine) with a harmonic potential of the form $V = 1/2k(x - x_0)^2$ with $k$ equal to 10.0 kcal/mol/Å$^2$. Molecular dynamics simulations were performed in YASARA (*Krieger & Vriend, 2015*) using the AMBER14 forcefield (*Maier et al., 2015*). Charge assignment was performed with the AM1BCC protocol (*Jakalian et al., 2000*; *Jakalian, Jack & Bayly, 2002*). Sampling was performed in bins 0.5 Å apart, for 6 ns per bin. In each bin, the first full ns was discarded from the analysis. The unbiased distributions were obtained through the weighted histogram analysis method (WHAM) (*Kumar et al., 1992*; *Grossfield, 2018*) using a bin size of 0.2 Å.
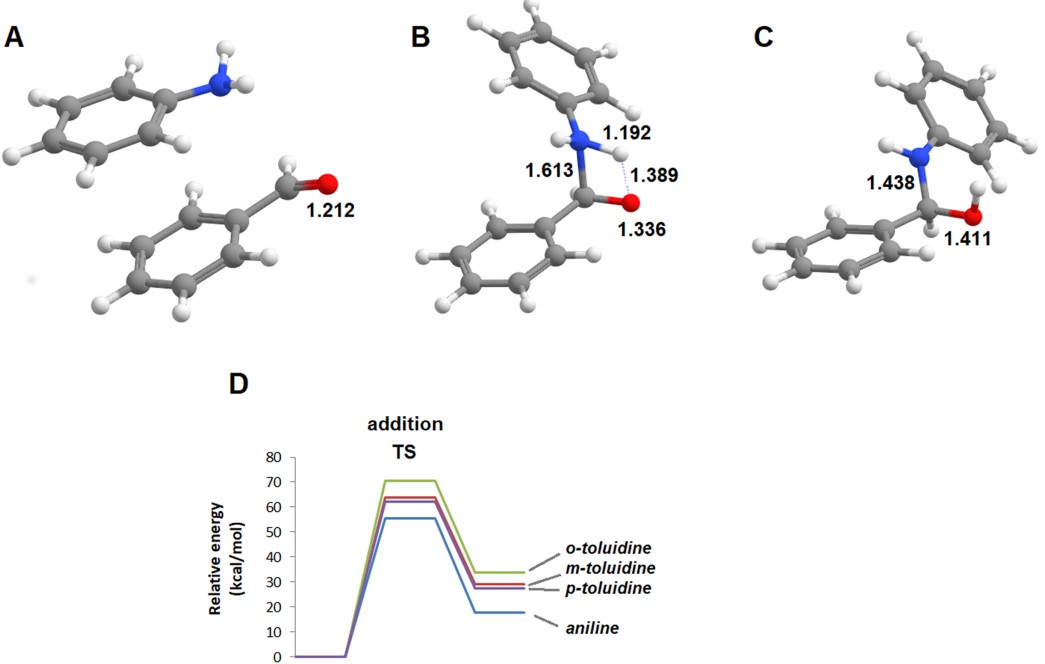

**Figure 2 Unassisted formation of carbinolamine.** (A) Reactants; (B) transition state; (C) carbinolamine; (D) reaction profile for the unassisted reaction of benzaldehyde with aniline (blue), *m*-toluidine (red), *o*-toluidine (green) and *p*-toluidine (violet).

## RESULTS

The reactions of benzaldehyde with aniline and its three mono-methylated derivatives (*o*-toluidine, *m*-toluidine, and *p*–toluidine) were studied in the gas phase. In all cases, the most stable initial arrangement of aldehyde and the aromatic amine finds both molecules parallel to each other due to the interaction between their aromatic clouds (Fig. 2). In the traditional description of this reaction mechanism, the subsequent formation of the carbinolamine intermediate proceeds through the simultaneous attack of the carbonyl carbon atom by the amine lone pair and proton transfer from the amine to the carbonyl oxygen atom. The geometry of this transition state (Fig. 2) is virtually identical for the four aromatic amines studied, with a N–C distance of 1.608–1.613 Å, a NH–O distance of 1.384–1.389 Å and a C–O distance of 1.336 Å almost exactly between that of a C–O double bond (1.215 Å) and a C–O single bond (1.411 Å). The transition states are, however, very hard to reach as they lie 55.4–70.6 kcal·mol$^{-1}$ above the pre-reactional complex state (Table 1). Since these high activation energies are incompatible with the experimentally observed syntheses of imines from aldehyde and aromatic amines at temperatures between 0 and 60 °C (*Allen & VanAllan, 1941*; *Campbell et al., 1948*; *Crowell & Peck, 1953*), the actual reaction mechanism must be more complex than commonly postulated.

Additional computations showed that the inclusion of an additional molecule of amine greatly facilitates the formation of the carbinolamine by assisting the proton transfer from the amine to the carbonyl oxygen (Fig. 3A). The increased energetic stabilization

**Table 1 Relative free energies vs. pre-reactional complex (kcal·mol⁻¹) of the species involved in the formation of carbinolamines from benzaldehyde and aniline derivatives.** Energies computed at the DSD-BLYP-D3(BJ)/pcseg2//PBE0-D3(BJ)/(aug)-pcseg1 theory level. Solvation effects in aniline were included with the PCM formalism.

|  | Aniline | *o*-toluidine | *m*-toluidine | *p*-toluidine |
|---|---|---|---|---|
| Transition state | 55.4 | 63.7 | 70.6 | 62.0 |
| Carbaminolamine | 17.8 | 29.1 | 33.8 | 27.5 |

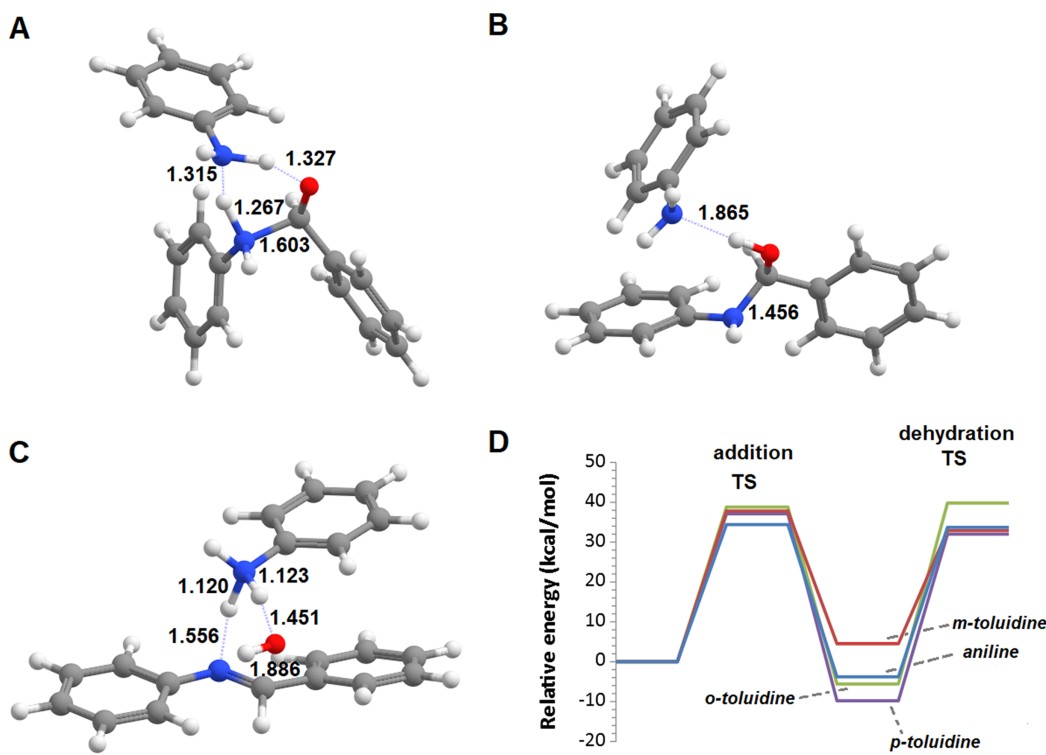

**Figure 3 Base-assisted formation of carbinolamine.** (A) First transition state; (B) carbinolamine and base; (C) transition state of the base-assisted dehydration; (D) reaction profile for the base-assisted reaction of benzaldehyde with aniline (blue), *m*-toluidine (red), *o*-toluidine (green) and *p*-toluidine (violet).

(between 21 and 32 kcal·mol⁻¹, relative to the unaided reaction) yields barriers of 34.4–38.8 kcal·mol⁻¹ above the pre-reactional complex state (Table 2), still far above the values required for detectable reaction rates. The reaction now becomes exergonic (vs. pre-reactional complex) in almost all cases, with the reaction with *o*-toluidine as the sole exception due to the geometric constraints entailed by the close proximity of the methyl substituents in the aromatic amines. Carbinolamine dehydration proved to be difficult (Table 2), with barriers ranging between 28 and 45 kcal·mol⁻¹ relative to the carbinolamine. Previous computational studies of this reaction (*Hall & Smith, 1998*; *Solís-Calero et al., 2012*) showed that this step is also difficult in water models unless large solvent cages are used, which allow very efficient charge delocalization throughout the hydrogen-bonded network (*Solís-Calero et al., 2012*). Since such stabilization is

**Table 2 Relative free energies vs. pre-reactional complex (kcal·mol$^{-1}$) of the species involved in the base-assisted formation of carbinolamines from benzaldehyde and aniline derivatives.** Energies computed at the DSD-BLYP-D3(BJ)/pcseg2//PBE0-D3(BJ)/(aug)-pcseg1 theory level. Solvation effects in aniline were included with the PCM formalism.

| | Aniline | *o*-toluidine | *m*-toluidine | *p*-toluidine |
|---|---|---|---|---|
| Transition state | 34.4 | 37.8 | 38.8 | 37.1 |
| Carbaminolamine | −3.8 | 4.5 | −5.6 | −9.8 |
| Dehydration transition state | 33.7 | 32.9 | 39.8 | 32.0 |

**Table 3 Relative free energies (kcal·mol$^{-1}$) of the species involved in the bimolecular dehydration of carbinolamines.** Energies computed at the DSD-BLYP-D3(BJ)/pcseg2//PBE0-D3(BJ)/(aug)-pcseg1 theory level. Solvation effects in aniline were included with the PCM formalism.

| | Aniline | *o*-toluidine | *m*-toluidine | *p*-toluidine |
|---|---|---|---|---|
| Two interacting carbinolamines | 0.0 | 0.0 | 0.0 | 0.0 |
| Bimolecular transition state | 21.0 | 17.7 | 5.2 | 18.3 |
| Carbinolamine + H$_2$O + product | 5.2 | −6.9 | −12.2 | −6.2 |

exceedingly unlikely to be available in aromatic amine solvents due to their inability to form such extended networks, we analyzed other possibilities of achieving acceptable reaction rates for the dehydration step. Inspired by the observation of dimeric derivatives of imines obtained through electrochemical reduction (*Law, 1912*) we evaluated the feasibility of stabilizing the carbinolamine dehydration step with a second molecule of carbinolamine. Interaction of two carbinolamines with each other to form a bimolecular pre-reactional complex is energetically very favorable (by 80–95 kcal·mol$^{-1}$, using DSD-BLYP(BJ)// PBE0-D3(BJ) in *implicit* solvent). Umbrella sampling simulations (Fig. S1) in a system containing two carbinolamines surrounded by 325 molecules of aniline confirmed that the formation of a pre-reaction complex composed of two mutually interacting carbinolamines with the correct relative orientations was indeed favorable in the presence of explicit solvent. The subsequent barriers range from 5.2 (for *m*-toluidine) to 18–21 kcal·mol$^{-1}$ (for the other aromatic amines) (Table 3; Fig. 4), well into the range of experimental feasibility, provided that a mechanism for the initial formation of the carbinolamine (that circumvents the high barriers encountered previously) can be found. The feasibility of acid-assisted catalysis was therefore explored. Our computations showed that the addition of one protonated molecule of amine greatly facilitated the attack of the carbonyl by the neutral amine, through the strong stabilization of the nascent negative charge on the carbonyl oxygen (Fig. 5). The N-protonated carbinolamine formed in this step may then transfer the extra proton to the assisting base, which in turn funnels it to the leaving hydroxyl group, yielding the protonated Schiff base and a water molecule. For the *aliphatic* amines tested, the results obtained were not very promising: although the initial formation of the N-protonated carbinolamine did indeed proceed without an energetic barrier, the subsequent proton transfer to the leaving hydroxyl group proved to be quite endergonic due to the relative

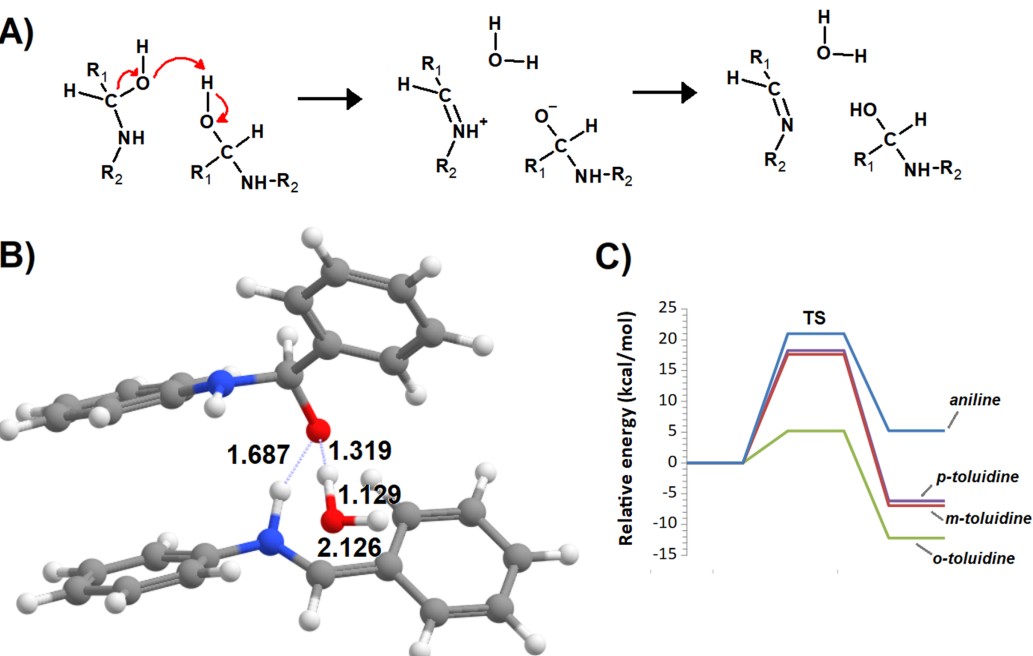

**Figure 4 Bimolecular dehydration of carbinolamines.** (A) Reaction scheme; (B) transition state; (C) potential energy surfaces of the bimolecular dehydration of the carbinolamines produced from the reaction of benzaldehyde with aniline (blue), *m*-toluidine (red), *o*-toluidine (green) and *p*-toluidine (violet).

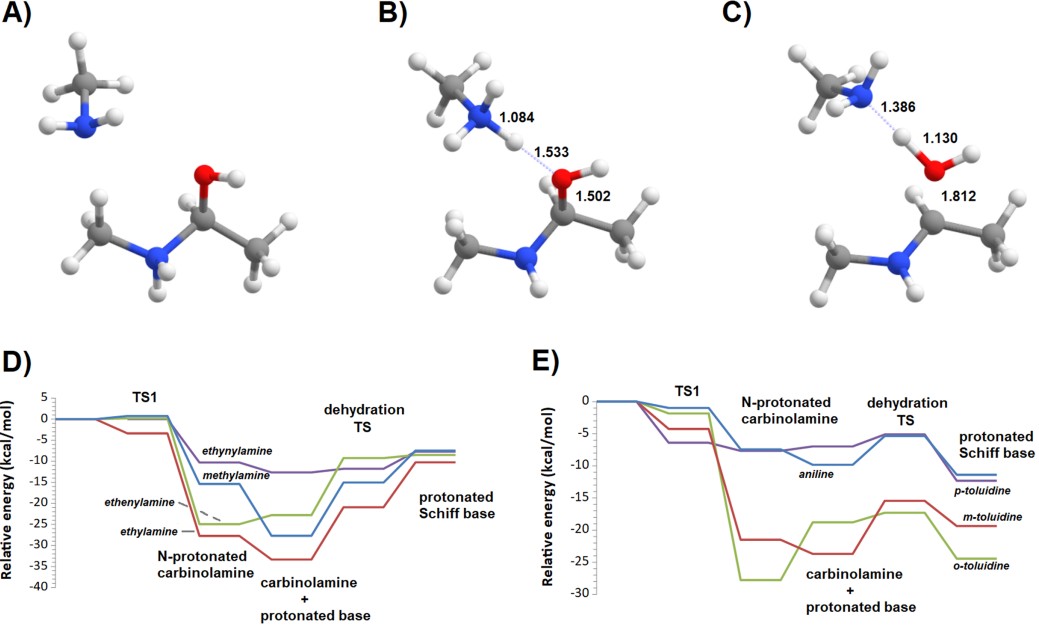

**Figure 5 Reaction profile of the protonated-amine-assisted formation of Schiff bases.** (A) N-protonated carbinolamine and neutral amine; (B) carbinolamine and protonated amine; (C) transition state of the dehydration step; (D) potential energy surfaces of the reactions of acetaldehyde with methylamine (blue), ethylamine (red), ethenylamine (green) and ethynylamine (violet); (E) potential energy surfaces of the reactions of acetaldehyde with aniline (blue), *m*-toluidine (red), *o*-toluidine (green) and *p*-toluidine (violet).

**Table 4 Relative free energies vs. pre-reactional complex (kcal·mol⁻¹) of the species involved in the acid-catalyzed synthesis of imines from acetaldehyde and different amines.** Energies computed at the DSD-BLYP-D3(BJ)/pcseg2//PBE0-D3(BJ)/(aug)-pcseg1 theory level. Solvation effects in aniline were included with the PCM formalism.

|  | Pre-reactional complex | TS1 | Int0 | Int1 | TS2 | Product | Highest barrier |
|---|---|---|---|---|---|---|---|
| Methylamine | 0.0 | 0.7 | −15.4 | −27.7 | −15.1 | −7.5 | 20.2 |
| Ethylamine | 0.0 | −3.4 | −27.7 | −33.3 | −20.9 | −10.3 | 23.1 |
| Ethenylamine | 0.0 | 0.2 | −24.9 | −22.8 | −9.3 | −8.5 | 16.4 |
| Ethynylamine |  | 0.0 | −10.3 | −12.7 | −11.8 | −7.7 | 4.9 |
| Aniline | 0.0 | −1.0 | −7.4 | −9.8 | −5.4 | −11.4 | 4.5 |
| *m*-toluidine | 0.0 | −1.8 | −27.8 | −18.8 | −17.3 | −24.4 | 10.5 |
| *o*-toluidine | 0.0 | −4.2 | −21.5 | −23.7 | −15.4 | −19.4 | 8.3 |
| *p*-toluidine | 0.0 | −6.4 | −7.7 | −7.0 | −5.1 | −12.3 | 2.5 |

**Table 5 Relative free energies (kcal·mol⁻¹) of the species involved in the aromatic aminium-assisted dehydration of the carbinolamines produced from the reaction of benzadelhyde with aromatic amines.** Energies computed at the DSD-BLYP-D3(BJ)/pcseg2//PBE0-D3(BJ)/(aug)-pcseg1 theory level. Solvation effects in aniline were included with the PCM formalism.

|  | Aniline | *o*-toluidine | *m*-toluidine | *p*-toluidine |
|---|---|---|---|---|
| Protonated carbinolamine + base | 0.0 | 0.0 | 0.0 | 0.0 |
| Carbinolamine + protonated base | 3.2 | 14.0 | 5.0 | 0.8 |
| Dehydration transition state | 7.1 | 19.4 | 14.6 | 8.6 |

instability of the produced N-protonated imine. Still, barriers of at most 23 kcal·mol⁻¹ (implying reaction rates of at least 0.3 h⁻¹) were obtained in all cases. The height of the barrier decreased markedly when the amine was changed from ethylamine to ethenylamine and ethynylamine, confirming that the introduction of π-delocalization stabilizes the product (and that the endergonicity of the reactions can be overcome by facilitating the spreading of the nascent positive charge throughout the molecule).

In agreement with this interpretation, the reaction barriers obtained (Tables 4 and 5) for the reactions of acetaldehyde or benzaldehyde with the four *aromatic* amines tested are very small (from 2.5 to 10.5 kcal·mol⁻¹) and therefore have extremely high reaction rates ($1.24 \times 10^5$ – $9.1 \times 10^{10}$ s⁻¹). The experimentally observed reaction rates (on the order of 1 h⁻¹) can therefore be achieved with minute concentrations of protonated base ($10^{-15}$ – $10^{-9}$ mol·dm⁻³).

We hypothesized that, even without the addition of acid catalysts, such minute amounts of protonated amine might be available through the auto-protolysis of the amine. Indeed, even some solvents generally regarded as aprotic or only weakly protic have been experimentally shown to auto-ionize to a limited extent (*Mihajlović et al., 1996*). To ascertain the likelihood of auto-protolysis of aniline and toluidines, we performed additional computations using small clusters of amine molecules, one of which was kept protonated (or deprotonated). Since very accurate results would require the simulation of

**Table 6 Computed auto-protolysis energies (kcal·mol$^{-1}$) of different solvents.** Geometries optimized at the PBE0-D3(BJ)/(aug)-pcseg1 level. Energies computed with DSD-BLYP-D3(BJ)/pcseg-2. Solvation effects were included with the PCM formalism. Experimental values were taken from *Mihajlović et al. (1996)*.

| | Auto-protolysis energies (kcal·mol$^{-1}$) | Experimental auto-protolysis constant |
|---|---|---|
| Ethenylamine | 69.4 | n.a. |
| Ethynylamine | 72.5 | n.a. |
| *p*-toluidine | 80.7 | n.a. |
| Aniline | 82.2 | n.a. |
| *o*-toluidine | 83.4 | n.a. |
| Ethylamine | 85.1 | n.a. |
| Ethylene carbonate | 88.2 | $10^{-21.5}$ |
| Acetonitrile | 95.8 | $10^{-28.8}$ |
| Methylamine | 100.9 | n.a. |
| Nitromethane | 105.9 | $10^{-23.7}$ |
| *m*-toluidine | 106.1 | n.a. |

very large solvent clusters to account for possible long-range structural rearrangements around the ionized structures, which are unfortunately not possible with our current computational resources, we compared our results with the auto-protolysis constants, computed in the same way, of other solvents which have been studied experimentally. Our results (Table 6) show that the auto-protolysis of most of the amines tested is much more favorable than that of ethylene carbonate (pKs = 21.5) nitromethane (pKs = 23.7) or acetonitrile (pKs = 28.8), and that therefore self-ionization of aniline or toluidines can easily afford concentrations of protonated amine at least as high as $10^{-21.5/2}$, which render accessible the mechanism postulated above.

## CONCLUSIONS

Like the analogous reaction in water (*Ding, Cui & Li, 2015*), imine formation from benzaldehyde and anilines in nonpolar solvent cannot occur without the intervention of a base which facilitates the transfer of one proton from the amine nitrogen atom to the carbonyl oxygen. Although the energetic stabilization provided by this assistance decreases activation energy by more than 20 kcal·mol$^{-1}$ relative to the reaction in the gas phase, this is not sufficient to enable reasonable rates of formation of the carbinolamine. The carbinolamine dehydration step is also prohibitively expensive, but can be made more accessible if a bimolecular mechanism (where one carbinolamine catalyzes the dehydration of the other) is taken into account. Both steps can be made much more accessible if the nascent negative charge in the attacked carbonyl (or the leaving hydroxyl in the dehydration step) are stabilized through interaction with the protonated forms of the reacting bases. Our computations, in turn, show that auto-protolysis of the amines to generate these species is feasible and that the low activation energies of the protonated amine-assisted mechanism fully enable the observation of good reaction rates even from the minute concentrations of protonated amine predicted to exist in water-free aniline/ benzaldehyde mixtures.

## ACKNOWLEDGEMENTS

This work was performed using computational resources acquired under a previous project (PTDC/QUI-QUI/111288/2009) financed by FEDER through Programa Operacional Factores de Competitividade–COMPETE and by Portuguese Funds through FCT–Fundação para a Ciência e a Tecnologia. The FP-ENAS Research Unit further receives some support from additional Portuguese Funds through a grant from FCT–Fundação para a Ciência e a Tecnologia (UID/Multi/04546/2019). Biosim is supported by the Applied Molecular Biosciences Unit-UCIBIO which is financed by national funds from FCT/MCTES (UID/Multi/04378/2019).

### Funding

The author received no funding for this work.

### Competing Interests

Pedro J. Silva is an Academic Editor for PeerJ.

### Author Contributions

- Pedro J. Silva conceived and designed the experiments, performed the experiments, analyzed the data, performed the computation work, prepared figures and/or tables, and approved the final draft.

### Data Availability

Data is available at Figshare:

Silva, Pedro (2019): Imine formation under catalysis by protonated amine: optimizations using (aug)-pcseg-1 basis set and D3-BJ, single points using pcseg2 and DSD-BLYP(D3-BJ). figshare. Dataset. DOI 10.6084/m9.figshare.9893987.

Silva, Pedro (2019): Imine formation from neat aromatic amines and benzaldehyde : optimizations using (aug)-pcseg-1 basis set and D3-BJ, single points using pcseg2 and DSD-BLYP(D3-BJ). figshare. Dataset. DOI 10.6084/m9.figshare.8794859.

### Supplemental Information

Supplemental information for this article can be found online at http://dx.doi.org/10.7717/peerj-ochem.4#supplemental-information.

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
