# Peer review of "New insights into the mechanism of Schiff base synthesis from aromatic amines in the absence of acid catalyst or polar solvents"

_PeerJ Organic Chemistry, doi:10.7717/peerj-ochem.4_

## Round 0.1 · original submission · Major Revisions

As you can see, we have received two rather short and one more substantial review response to your article. In summary, I agree with the points raised by the reviewers and think revisions are needed for the article to be published.

Reviewer 1 ·

Basic reporting

This article appears to meet the stated standards.

Experimental design

This article appears to meet the stated standards.

Validity of the findings

I have concerns about the results. While the calculations were rather straightforward, the conclusions seem to be drawn on particular orientations of a single water molecule. To my mind, the authors would need to sample a multitude of possible orientations to assure that configuration space is adequately covered to draw meaningful conclusions. In addition, I worry about carrying out such sampling in the presence of a continuum rather than a large number of explicit solvent molecules.

·

Basic reporting

Manuscript is very well written. I suggest addition of discussion to the introduction regarding importance of understanding Schiff base formation. I would also suggest substituting "apolar" for "nonpolar" throughout.

Experimental design

Methods are likely to be adequate. However, I suggest the authors add discussion on the appropriateness of the PBE0 method for such systems.

Validity of the findings

All data provided are appropriate.

Reviewer 3 ·

Basic reporting

This is a purely computational study investigating pathways towards imine synthesis with specific emphasis on solvent effects. Prior work has been referenced appropriately. I have no problem with this being an entirely computational study without any experimental evidence, however, as outlined further below, I do have some concerns regarding the methodology that has been used and the reliability of the results. I also find the discussion relatively short and feel the author could elaborate more on the findings. A few things, which I will mention below, are not quite clear and hence it would be difficult to reproduce the results. I think a major revision is warranted.

Experimental design

The positive thing is that the author took into account London-dispersion interactions, something that has been often ignored in computational organic chemistry in the past. That being said, it is surprising that the author uses an outdated version of the DFT-D3 correction instead of the more reliable and accurate 2011 version with Becke-Johnson damping. There is also no reason given why PBE0 was chosen as the functional (more on this in the next point). The basis sets seem to be OK. The author mentions explicit solvation, but even after rereading various sections of the manuscript multiple times, I would not be able to reproduce those results if I wanted to. How many explicit solvent molecules were included? Only one? If that is the case, this would not qualify as a reliable study.

Validity of the findings

1)I already commented on how difficult it was to understand how exactly solvent molecules were treated explicitly? The author does not mention any sampling of the solvent shell, how many molecules were included etc.

2) The discrepancy between experiments and calculated high activation barriers could not only be due to the postulated reaction mechanism (see ll. 71-74), but due to the chosen methodology. However, such critical analysis is missing. In fact, The reliance on PBE0 is a bit worrying. Major benchmark studies have shown that it is only a mediocre functional for reaction energies and barrier heights (see the GMTKN55 and MGCDB84 databases and subsequent recommendations and insights). While I can accept this functional for geometry optimisations, the systems (at least in the gas phase) are not prohibitively large for dispersion corrected double hybrids or at least the better hybrid functionals recommended in the aforementioned studies. As such, I do not agree that PBE0-D3/aug-pcseg-2 can be considered a “high level” (line 49). If the author decides to stick with the chosen methodology, then a paragraph is needed that puts it into context with larger benchmark studies. An expected error range should be provided and it should be made clear that this is only a very qualitative study.

3) What type charges are reported in l. 78 and the lines thereafter? Mulliken charges or more reliable ones?

---

## Round 0.2 · Minor Revisions

I follow the judgment of the referees that the manuscript addresses all previous concerns, which leaves only minor revisions to be conducted.

Reviewer 1 ·

Basic reporting

I would add a brief description of the MD test on solvent mentioned in the response letter.

Experimental design

I still have concerns about using single solvent molecules but I am okay with readers deciding whether or not they believe the results. It is at least clear what was done.

Validity of the findings

See point #2.

Reviewer 3 ·

Basic reporting

This manuscript has addressed all of my concerns and there are only two minor things for a very minor revision.

Experimental design

The computational strategy has improved. Choosing a double hybrid was a very good move to give the study more credibility. What is not clear is why the first sentence in the Computational Methods section reads that PBE0 was used to investigate the reaction mechanism if the final energies were based on DSD-BLYP? Maybe it would be better to say that a combination of two levels of theory was used to explore the reaction mechanism? PBE0-D3(BJ) for the geometries and thermal corrections, DSD-BLYP-D3(BJ) for the electronic energies?

Validity of the findings

The findings appear to be valid

Additional comments

PBE0 was published around the same time as PBE1PBE by Scuseria and co-workers, which is exactly the same functional. It is common practice to cite both papers.

---

## Round 0.3 · accepted · Accept

I am happy to accept the manuscript after the last round of minor revisions and congratulate you for the paper